# Control of Adipose Cell Browning and Its Therapeutic Potential

**DOI:** 10.3390/metabo10110471

**Published:** 2020-11-19

**Authors:** Fernando Lizcano, Felipe Arroyave

**Affiliations:** 1Center of Biomedical Investigation, (CIBUS), Universidad de La Sabana, 250008 Chia, Colombia; 2Doctoral Program in Biociencias, Universidad de La Sabana, 250008 Chia, Colombiafelipe.arroyave@unisabana.edu.co

**Keywords:** beige adipocyte, white adipocyte, brown adipocyte, obesity, diabetes mellitus, differentiation

## Abstract

Adipose tissue is the largest endocrine organ in humans and has an important influence on many physiological processes throughout life. An increasing number of studies have described the different phenotypic characteristics of fat cells in adults. Perhaps one of the most important properties of fat cells is their ability to adapt to different environmental and nutritional conditions. Hypothalamic neural circuits receive peripheral signals from temperature, physical activity or nutrients and stimulate the metabolism of white fat cells. During this process, changes in lipid inclusion occur, and the number of mitochondria increases, giving these cells functional properties similar to those of brown fat cells. Recently, beige fat cells have been studied for their potential role in the regulation of obesity and insulin resistance. In this context, it is important to understand the embryonic origin of beige adipocytes, the response of adipocyte to environmental changes or modifications within the body and their ability to transdifferentiate to elucidate the roles of these cells for their potential use in therapeutic strategies for obesity and metabolic diseases. In this review, we discuss the origins of the different fat cells and the possible therapeutic properties of beige fat cells.

## 1. Introduction

Obesity is a disease that induces a series of cardiovascular, metabolic and osteoarticular complications that reduce life expectancy. It is prevalent on a global scale, with multiple factors contributing to its development. The treatment possibilities for metabolic diseases have increased dramatically. However, in the circumstance of obesity, many medications have been withdrawn from the market due to undesirable side effects [1,2]. Other drugs have not presented the desired efficacy and the projections of therapeutic efficiency are low with high costs. Furthermore, one of the difficulties in many countries is that obesity has not been declared a disease. Obesity is a highly stigmatized condition that has long been generally regarded by the public as a reversible consequence of personal choices. For this reason, obesity is seen even in some countries, as a circumstance of the person, without policies for its prevention, adequate therapy and of course the risk of the appearance of complications is not supported [3,4]. Recent observations have proven that a variety of types of adipose tissue dysfunction clearly play a role in the genesis of many obesity-related diseases. These include impairments in adipocyte storage and release of fatty acids, overproduction or underproduction of “adipokines” and cytokines, hormonal conversion, and the adverse mechanical effects of greater tissue mass [5,6]. Additionally, adipose tissue has been shown to be dynamically more active than initially considered [7]. Adipose tissue is constituted by white adipose tissue (WAT), which has the property of storing energy in the form of triglycerides and is useful in preventing deficiencies of energy during periods of prolonged starvation. In comparison, there is the brown adipose tissue (BAT), which is more metabolically active and has the property of producing heat through the activation of uncoupling proteins (UCP1). BAT controls energy homeostasis during periods of low temperature and hibernation [8]. In adult humans, WAT is believed to predominate, while BAT has a predominant role in the first few months of life when heat production by adipose tissue is necessary to maintain body temperature [9]. However, another type of adipose tissue has been described in adults, whose functional characteristics may be similar to those observed in BAT. This type of adipose tissue is observed in circumstances of low temperatures or after sympathetic activation [10]. Although this adipose tissue is functionally very similar to BAT, it has specific characteristics and for this reason it has been referred to as beige adipose tissue [11,12]. Although adipose tissue has a mesenchymal origin, in the differentiation process manifest differences are established and BAT has more similarities with muscle cells of mesenchymal origin than with WAT [13,14]. Additionally, in the process of differentiation, some external factors can modify the cells that give rise to WAT and change the phenotype of these cells towards cells that are more metabolically active, such as beige adipocyte [15,16]. In the human body, energy homeostasis is regulated in the central nervous system by the hypothalamus, which has several widely interconnected hypothalamic neural circuits. The hypothalamus is involved in the physiological control of many functions, including metabolic homeostasis. The chemical signals that can modulate the function of the hypothalamus are the signals that come from the immune system, those that come from the activation of the sympathetic nervous system and other signals that can induce epigenetic modifications, which influence the expression of specific genes that influence the transcription of specific genes [17,18]. This review discusses the metabolic control of organisms, the effect of the environment on the regulation of thermogenesis and the different adipocytes identified to date, emphasizing the therapeutic potential of beige adipocytes.

## 2. Hypothalamic Control of Energy Homeostasis

The control of energy in the body is based on the balance between the intake of energy through food and the expenditure of calories. Energy homeostasis is regulated in the central nervous system by the hypothalamus, which has several widely interconnected hypothalamic neural circuits. The hypothalamus is involved in the physiological control of many functions, including the regulation of hormonal axes, autonomic nervous system activity, and metabolic homeostasis [19]. Hypothalamic nuclei play a major role in the transmission of information from peripheral signals on energy availability, including hormone and nutrient signals, integrating them and generating an appropriate response in terms of food intake and energy expenditure [20,21]. Due to its involvement in numerous metabolic processes, the hypothalamus is considered to be the master regulator of energy homeostasis [22].

Energy expenditure is the sum of the thermic effect of food, locomotor activity and thermogenesis [23,24]. Interestingly, basal thermogenesis, which is the heat produced by metabolism, is sufficient to preserve body temperature at adequate levels without involving thermoregulatory mechanisms. This range of temperatures in which the body is maintained in a harmonic state with the environmental temperature is called thermoneutrality [25,26], with temperatures below thermoneutrality inducing an immediate response through peripheral vasoconstriction [24]. However, this primary response only provides limited effects in maintaining body temperature. Therefore, the body uses additional thermogenic mechanisms, referred to as adaptive thermogenesis [27], that can be induced by shivering or involve non-shivering thermogenesis. Whereas shivering produces additional heat from movement, mammals have developed a non-shivering adaptive thermogenic mechanism that is carried out by BAT [28]. Core temperature control is essential to preserve energy homeostasis. The preoptic area (POA) is located in the anterior region of the hypothalamus and controls body temperature. The POA has the ability to receive thermosensitive peripheral signals from the skin and intestinal organs and trigger the activation of efferent signals that can promote BAT thermogenesis [29,30].

Moreover, changes in behavior, including the waking state, the response of the immune system and stress are characterized by elevated body temperature. Although the neural circuits and transmitters involved in the response to behavioral changes are not well understood, the activity of orexins may have an important role in these behavioral events [31]. Additionally, the high metabolic rate of beige adipose tissue and BAT during thermogenesis cannot be maintained without a reliable supply of metabolic fuels, particularly oxygen, lipolytic products, and glucose [32,33]. A great deal of information has been obtained from the study of neuronal regulators, specifically, those that function at the hypothalamic level that are responsible for these events. The arcuate nucleus (ARC) of the hypothalamus is the region that is most implicated in the control of eating habits. The ARC is made up of two primary neural populations: (a) orexigenic (feeding-promoting) neurons and co-expressing neuropeptide Y (NPY) and related protein agouti (AgRP) and (b) anorexigenic (feeding-inhibiting) neurons that co-express a transcription factor related to cocaine and amphetamines (CART) and pro-opiomelanocortin (POMC), the precursor of alpha-melanocyte-stimulating hormone (α-MSH) and adrenocorticotrophic hormone (ACTH) [22,34]. After integration by the ARC, peripheral signals are transmitted by neural projections to hypothalamic areas of the dorsomedial nucleus (DMH), paraventricular nucleus (HPV), lateral hypothalamic area (LHA), and ventromedial nucleus of the hypothalamus (VMH) [35]. VMH neurons can communicate with other hypothalamic areas, such as the DMH, LHA, and ARC, in addition to other brain regions, such as the vagus motor dorsal nucleus (DMV), the nucleus of the solitary tract (NTS), the pale raphe (RPa) and the lower olive (IO) [36].

In addition to the established biochemical neurotransmission mechanisms, functional studies have identified specific areas of the brain that generate WAT browning. For example, the role of neuropeptide-Y (NPY) in DMH nuclei, in addition to oxidative stress and the administration of CART to the paraventricular nucleus (PVN), has been shown to induce an increase in uncoupling protein1 (UCP1) levels in WAT [37]. The regulation of WAT metabolic activity by thyroid hormones, bone morphogenic protein 8B (BMP8b) and the incretin glucagon-like peptide-1 (GLP-1) can be reduced by blocking AMP activated protein kinase (AMPK) in the VMH nucleus [28]. Increased expression of the endoplasmic reticulum (ER) chaperone protein GRP78, which improves ER stress, induces signal activation mediated by β-3-adrenergic receptors, increasing browning and reducing weight [38].

In recent years, several studies have shown the important role that the stimulation of LHA nucleus neurons has in BAT activation. The nerve terminals that reach beige adipocytes promote the browning process. Under cold exposure, the central neural circuits in the hypothalamic (HPV and LHA) and brain stem of RPa and locus coeruleus are rearranged with higher proportions of neurons projecting into BAT and beige adipocytes [39]. These data provide strong evidence indicating a likely reorganization of nervous system connectivity after WAT browning [40] (Figure 1).

The results of pharmacological experiments and animal studies have shown that orexins induce increased energy expenditure. Mice with reduced orexin expression show a reduced ability to maintain temperature after exposure to cold [41,42]. The administration of orexin A and B to the VMH and LHA nuclei of rats induces the thermogenic activity of BAT [43,44,45]. There is evidence to indicate that orexins can induce the thermogenic activity of BAT by regulating the energy sensor AMPK and ER stress [46]. Some groups have observed that BMP8b can affect both BAT and the browning process of WAT [42,47]. Despite multiple lines of evidence showing the role of orexins in rodents, it has not been possible to elucidate their roles in stimulating BAT activity in humans. Treatment with Orexin A alone or in combination with an adrenergic stimulation does not affect thermogenesis [48,49]. These observations are closely related to findings in patients with narcolepsy, a disease that involves the selective deterioration of neurons that produce orexin. Patients with narcolepsy present an abnormal distribution of body fat but retain BAT deposits at the supraclavicular level [50]. Additionally, BAT has been shown to be functional in these patients after cold exposure. These observations show that the role of orexins in humans is controversial and that the control of thermogenesis and fat cell activity at the hypothalamic level in humans requires additional studies [51]. In humans, the control of thermogenesis may be influenced by multiple hormones that are secreted in different tissues, such as insulin and glucagon by the pancreas; leptin by adipose tissue; incretins, gastric inhibitory peptide (GIP), GLP-1, and ghrelin produced in gastric fundus of the gastrointestinal system; thyroid hormones and estrogens. All of these hormones may affect the regulation of adaptive thermogenesis in humans [52,53,54,55].

Energy consumption and food composition can influence the thermogenic activity of BAT. In addition to the heat released during the digestive process and nutrient transport, food has a postprandial thermal effect. Glucose and insulin can contribute to the activation of the sympathetic system and b-adrenergic receptors [56]. However, the activation of thermogenesis by glucose may be influenced by other hormones such as cholecystokinin (CCK), which acts as a sensor of the caloric value of other nutrients [57]. Hypoglycemia induces hypothermia at least in part by reducing the thermogenic activity of BAT. The reduction in glucose in the VLM nuclei completely inhibits BAT activity [58]. The contribution of lipids in the activity of postprandial thermogenesis is characterized by the activation of sympathetic nerve activity. In rodents, diets with a high lipid content stimulate the thermogenic activity of BAT and the activation of b-adrenergic receptors [59]. However, this effect diminishes after a few weeks [60]. Other nutrients that can stimulate thermogenesis are proteins. Although the effect of proteins on BAT is not yet fully elucidated, ketogenic diets with a high content of lipids and proteins and no carbohydrates induce activation of the sympathetic system [61]. High levels of ketone bodies increase BAT and energy expenditure. β-Hydroxybutyrate increases BAT activity and norepinephrine secretion, and the most likely mechanism for the effect of (beta) β-Hydroxybutyrate is through the VMH and HPV [62].

## 3. Environment and Obesity

Endocrine-disrupting chemicals (EDCs) are substances that are located in the environment and interfere with normal hormone action, thereby increasing the risk of developing diabetes, cancer, reproductive impairment, behavior disorders and obesity. These chemical substances have been shown to potentially cause alterations in metabolic activity in organisms and increase lipogenic activity [63]. Adipocytes are functionally classified as endocrine-active and are sensitive to changes induced by endocrine disruptors, which have been termed “obesogens” [64]. These substances may trigger an increase in adiposity by altering the cellular development program in adipocytes, increasing triglyceride storage and interfering with the neuroendocrine control of hunger and satiety centers [65,66]. Some of the chemicals we typically encountered tend to cause changes in fat cell metabolism. Bisphenol A (BPA), phytoestrogens, and tributyltin (TBT), among others, have been shown in experimental studies to induce weight increases in animal models. Although there are multiple mechanisms by which endocrine disruptors can induce obesity, the modifications described so far suggest that epigenetic modifications of gene expression through DNA methylation and covalent modifications of chromatin are common and affect subsequent generations [67,68].

Numerous potentially obesogenic compounds have been identified using in vitro assays that evaluate the ability of candidate chemicals to promote the differentiation of established cell lines, such as mesenchymal stem cells from humans and mice, and 3T3-L1 preadipocytes [69,70]. Many of the chemicals shown to promote the differentiation of white adipocytes in these analyses activate PPARγ and/or RXR [71]. Based on the central role of the PPARγ:RXR heterodimer as the master regulator of adipogenesis, it is not surprising that these nuclear transcription factors are one of the most important targets of obesogens [72,73,74,75] (Figure 2).

Human exposure to organotin can occur through dietary intake, such as by seafood contaminated with TBT, or after the consumption of products that have been in contact with pesticides such as triphenyltin [76,77].

TBT has the ability to bind and activate PPARg, promoting adipogenesis and lipid accumulation. The results of studies using 3T3-L1 cells and human mesenchymal cells have shown that TBT at nanomolar concentrations can induce their differentiation into adipocytes [78]. Studies in humans have revealed that elevated levels of TBT in urine are associated with metabolic diseases such as diabetes and obesity. In a recent study, individuals with high levels of perfluorinated chemicals exhibited a low metabolic rate and tended to gain weight easily [79].

Similar observations have been made with other EDCs, such as phthalates, plastic components, and epoxy resins. The phthalate MEHP (mono-2-ethylhexyl phthalate) can induce adipogenesis in 3T3-L1 cells by activating PPARγ. Prenatal exposure to bisphenol A (BPA) has been linked to several adverse effects, including weight gain, reproductive disorders, and behavioral changes in mice and rats. BPA can interfere with the activity of estrogen receptors (ERs) and with PPARγ [80].

Although several studies have observed that obesogens primarily influence PPARγ receptor activity, recent studies have shown that other functional variations can be attributed to obesogens, such as modifications to the retinoid acid receptor and other nuclear receptors, such as glucocorticoid or thyroid hormone receptors [81,82]. Other functional variations have been observed after epigenetic modifications in WAT, including structural changes in chromatin or modifications in the gut microbiota [83,84].

Epigenetic modifications mediated by TBT include a reduction in the trimethylation of histone 3 lysine 27 (H3K27me3), which increases the expression of genes involved in the development of adipogenesis [85]. Other modifications have also been observed, such as a reduction in DNA methylation [86], and it has even been argued that DNA methylation modifications may have a transgenerational impact [87,88]. However, both the evidence and the mechanisms for this epigenetic modification have not been fully elucidated. In the induction of adipogenesis by TBT, these adipocytes have been shown to exhibit an altered functionality due to changes in oxidative respiration, difficulty in expressing thermogenic proteins after cold stimulation or activation of b-adrenergic receptors. These adipocytes also tend to be insulin resistant and express profibrotic proteins [89,90].

Recently, other chemicals with potential obesogenic effects have been described. The TBT metabolite dibutyltin (DBT), the prevalence of which is higher than TBT in the environment, can also induce 3T3-L1 preadipocyte differentiation. In mouse studies, DBT has been observed to cause insulin resistance [91]. Analogs of bisphenol A, bisphenol S (BPS) and bisphenol F (BPF) are able to activate PPARγ and induce adipogenesis. A longitudinal study of a cohort of children demonstrated that exposure to BPS and BPF was significantly related to obesity in children [92,93]. Acrylamide is widely used as a colorant in the manufacture of paper and other industrial products and can be generated after cooking foods with a high carbohydrate content at high temperatures. Interestingly, acrylamide increases the activity of the adenosine 5′-monophosphate-activated protein kinase-acetyl-CoA carboxylase (AMPK-ACC) pathway [94]. The results of two European studies, one in France and one in Norway, showed that children exposed to high levels of polyacrylamide during the prenatal period were more likely to be short for their gestational age and tended to be obese after 3 years of age [95,96]. Surfactants such as dioctyl sodium sulfosuccinate (DOOS) and Span 80 can activate the PPARγ and RXR receptors, and the combination of these products can induce an increase in adipogenesis. Other chemical compounds used as preservatives, such as 3-tertbutyl-4-hydroxyanisole (3-BHA), induce the differentiation of 3T3-L1 preadipocytes [97], while the flavoring monosodium glutamate (MSG) can promote adipogenesis by modifying the secretion of GLP-1. In addition, some herbicides that remain in use in some countries, such as glyphosate, can induce obesity in the F2 and F3 offspring of females exposed to during gestation [98].

## 4. Functional Variation of Adipocytes

Adipose tissue has allowed mammals to adapt to changes due to energy demand, environmental conditions, and nutrient availability. In recent years, knowledge of adipose tissue has been ostensibly expanded, which is partly due to increases in metabolic diseases and cardiovascular risk [99,100,101]. Although fat cells are the primary component of adipose tissue, almost 40% comprises vascular components, macrophages, fibroblasts, endothelial cells and adipocyte precursor cells [89,102,103]. The size of the adipocytes can vary considerably from 20 to 200 µm in diameter, showing that they have great plasticity and a high capacity to modify their volume [104,105]. Adipose tissue pathologies can result as a consequence of lipid accumulation and adipocyte cell hypertrophy. WAT secretes endocrine and inflammatory signals under hypertrophic conditions, inducing a prothrombotic state and generating a chronic inflammation disorder [106,107]. This condition causes many of the cardiovascular complications presented by patients with metabolic diseases [108,109]. Most of the adipose tissue in adults is WAT, which has the functional characteristic of saving energy for periods of famine. Phenotypically, white adipocytes have a single cytoplasmic lipid droplet that is responsible for storing triglycerides as a consequence of lipogenic processes. Recent observations have shown that WAT exhibits great phenotypic plasticity [7,110]. These adipocytes, under the regulation of the sympathetic nervous system, can release fatty acids and secrete substances with endocrine effects [111]. Since the discovery of leptin in the 1990s [112,113], adipose tissue has not only been seen as an exclusive energy storage organ but also as a dynamic organ with an endocrine function, and several WAT-secreted adipokines with various biological activities have been described in recent years. Leptin regulates the energy homeostasis of the body and interferes with various neuroendocrine and immune functions, regulating food intake and increasing energy utilization through hypothalamic signals [114]. Leptin administration reduces the weight of mice or humans with congenital leptin deficiency [115]. However, the function of leptin in humans with diet-induced obesity is subject to the control of receptors and transporters that make pharmacological therapy difficult [116,117]. Adiponectin is another adipokine that is predominantly secreted by WAT, although it can also be secreted by skeletal muscle and cardiomyocytes. Adiponectin plays a crucial role in glucose and lipid metabolism, inflammation and oxidative stress. Adiponectin levels increase with exposure to insulin-sensitizing drugs, with adiponectin plasma levels observed that are inversely proportional to insulin resistance. Adiponectin also has anti-inflammatory properties with anti-atherogenic effects and promotes angiogenesis [118,119]. Resistin is secreted by WAT and macrophages and has an important role in inflammatory processes that trigger insulin resistance, with some studies having determined that elevated plasma levels of resistin are a predictor of the future development of type 2 diabetes mellitus [120,121]. The mechanisms by which resistin can induce insulin resistance are not fully elucidated in humans. However, the results of in vitro studies have shown that the activities of pro-inflammatory cytokines such as tumor necrosis factor-α (TNF-α) and interleukin 6 (IL-6) in addition to the functional modification of 5′AMP-activated protein kinase (AMPK) may be involved in resistin-mediated insulin resistance [122]. Visfatin, also known as nicotinamide phosphoribosyl-transferase (Nampt), is an adipocytokine secreted by adipocytes, macrophages, and inflamed endothelial tissue. High levels of visfatin are observed in patients with obesity, type 2 diabetes mellitus, chronic inflammatory conditions and cancer, and an association between serum visfatin levels and cardiovascular disease has recently been observed in patients with type 2 diabetes [123,124,125]. Moreover, BAT exhibits multiple cytoplasmic lipid inclusions and numerous mitochondria. Compared to WAT, BAT is highly vascularized and rapidly metabolizes fatty acids, favoring optimal oxygen consumption and heat production [23]. Many environmental or molecular stimuli can increase the appearance of BAT [126]. Brown adipocytes are primarily observed in small mammals and in the newborn, with the embryological formation of BAT preceding that of WAT due to its thermogenic function in newborns. BAT originates from a subpopulation of the dermomyotome that expresses molecular markers such as paired box 7 (Pax7), engreiled-1 (En1), and myogenic factor 5 (Myf5) [7,127,128,129]. BAT can secrete cytokines that have an effect on different tissues and prevent diet-induced obesity. Follistatin is a soluble glycoprotein secreted by BAT that can blockade the activities of some members of the transforming growth factor (TGF) family, induce insulin sensitivity and prevent diet-induced obesity [130,131]. The c-terminal fragment of slit guidance ligand 2 (SLIT-C) belongs to the Slit family of secreted proteins that play an important role in various physiological and pathological activities, including inflammatory cell chemotaxis. SLIT-C is secreted by BAT and induces thermogenic WAT browning and metabolic processes associated with substrate supply to fuel thermogenesis [128,132]. Growth differentiation factor 8 (GDF8, also known as myostatin) and growth differentiation factor 15 (GDF15) are members of the transforming growth factor family, which are involved in the control of hunger-related neural circuits. GDF15 overexpression has been shown to prevent obesity and insulin resistance by increasing the expression of thermogenic genes [133]. Fibroblast growth factor 21 (FGF21) is a regulator of energy homeostasis that is primarily secreted by the liver. BAT-secreted FGF21 prevents hyperglycemia and hyperlipidemia in mice [134], and FGF21 analogues tested in overweight/obese patients with type 2 diabetes mellitus have been shown to reduce dyslipidemia and hepatic steatosis, although they do not lead to improvements in glucose control and body weight [135]. Although FGF21 was reported to have anti-inflammatory effects on white adipocytes, it remains to be determined if FGF21 has a similar action in BAT [136,137].

Beige adipose tissue is the newest of these adipose tissues and has some morphological characteristics in common with WAT and BAT. The nature of these cells is controversial, although it is believed that their origin is secondary to the differentiation of WAT, and their differentiation from cell precursors has been observed [30,138]. Beige adipocytes have simple lipid inclusions similar to WAT, but when faced with stimuli such as cold exposure, their behavior is similar to that of BAT cells. The thermogenic capacity and potential role of beige adipose tissue in the regulation of obesity and insulin resistance are currently being studied [139,140]. Beige adipocyte biogenesis, also called beige adipogenesis or (browning/beigeing), is induced by chronic exposure to external cues such as cold, adrenergic stimulation, and long-term treatment with peroxisome proliferator-activated receptor gamma (PPARγ) agonists, among others [141] (Table 1).

Browning is a temporary adaptive response that lasts even after the dissipation of external environmental signals [140,142]. Beige adipocytes have an origin that is not yet fully clarified. Some are believed to arise from WAT from cell precursors that express CD34, in addition to platelet-derived growth factor receptor alpha (PDGFRα), and spinocerebellar ataxia type 1 (SCA1) proteins [143,144,145]. Beige adipocytes may also be derived from the Myf5-negative precursors of inguinal WAT [30], and the results of a number of studies suggest that beige adipocyte precursors, such as WAT precursors, reside in adipose tissue vasculature [141,146,147]. Recently, some beige adipocytes have been shown to express myosin heavy chain 11 (Myh11), which is a selective marker of smooth muscle cells [141]. These observations may indicate that during embryonic development beige adipocytes have a different cellular origin from that seen with classic brown adipocytes [148,149,150].

Another of the functional characteristics of beige adipocytes in relation to other adipocytes is their functional flexibility. Although the differentiation process of beige adipocytes from precursor cells is highly inducible, there is clear evidence that mature white adipocytes could be transdifferentiated into beige adipocytes by specific exogenous factors [105].

Whether this change is manifestation of a real transdifferentiation from white adipocytes, a direct transformation of white adipocytes to beige adipocytes, or resembles beige adipocytes that previously remained hidden among white adipocytes is a matter of debate [151]. One of the considerations that are still under evaluation is the amount of BAT in adult humans, because it is considered that most of the thermogenic adipose tissue in adults corresponds to beige adipocytes [148,152,153,154]. However, BAT can be observed in adults in specific areas, such as the posterior neck and perirenal area [155,156,157,158]. The adipose progenitor cells (APC) maintain a continuous supply of adipocytes in the different tissues in the body. In this way, the number of adipocytes in the body remains constant in adults despite the fact that the individual is obese or thin. This indicates that the amount of adipocytes in the body is established during childhood and adolescence in a correspondent way with the size of the different organs of the body [150]. According to recent studies, adult APC cells are considered to reside in the stromal vascular fraction (SVF). Specifically, studies in mice have identified cells that have APC characteristics in SVF and express PPARγ [149,159]. Using genetic-tracing methodologies, PPARγ-expressing APCs were shown to be crucial for adipocyte formation in vitro and in vivo [160]. In vivo tracking of PPARγ cells has indicated that these cells reside within blood vessel walls. In line with a vascular residency, these APCs resemble mural cells (pericytes and vascular smooth muscle cells) due to their expression of several mural cell markers, such as platelet-derived growth factor receptor-beta (PDGFRβ) and alpha-smooth muscle actin (α-SMA). The results of smooth muscle genetic fate-mapping studies have suggested that cells expressing Myh11, PDGFRβ, and SMA can generate beige adipocytes in response to cold exposure [12,161]. SMA perivascular cells have been shown to generate 50–70% of new beige adipocytes after 1 week of cold exposure. Remarkably, blocking adipogenesis within SMA cells or ablating SMA positive cells led to the failure to generate cold-induced beige adipocytes, and mice were unable to either preserve their temperature or lower plasma glucose levels [162,163].

## 5. Thermogenesis by Brown and Beige Adipose Tissue

Free energy for life-sustaining biochemical processes in mammals is provided by reduced substrates. The energy in the cells is subject to the oxidation of substrates in the inner membrane of the mitochondria through oxidative phosphorylation. In this electrochemical pathway, proton conductance is established by the mitochondrial respiratory chain producing energy [164]. Thus, cell metabolism is carried out in the membranes of the mitochondria through oxidative phosphorylation. The protonmotive force (Dp) generated by mitochondrial respiration drives protons back into the mitochondrial matrix through ATP synthase, providing energy for the reaction ADP+Pi/ATP [165]. The hydrolysis of ATP into ADP and inorganic phosphate releases 30.5 kJ / mol, with a change in free energy of 3.4 kJ/mol [166]. The energy released by the division of a unit of phosphate (Pi) or pyrophosphate (PPi) of ATP in the standard 1 M state is: ATP + H_2_O → ADP + Pi ΔG = −30.5 kJ/mol (−7.3 kcal/mol). Interestingly, most of the thermal energy that is produced from the oxidation of substrates is conserved in a small fraction, and most of this energy is released in the form of heat. Thereby, in cells that maintain oxidative metabolism from the mitochondrial respiratory chain, the production of heat is subject to the rate of mitochondrial respiration. Thermogenesis in brown and beige adipocytes is effectively controlled by modulating the steps that modify the mitochondrial respiratory chain [167,168]. UCP1, previously referred to as thermogenin, is responsible for the conductance of protons in brown and beige adipocytes. Taking into account the laws of thermodynamics, most of the energy generated by the electrochemical potential in the oxidation of brown and beige adipocytes is dissipated as heat and is not used for the phosphorylation of ADP. Thus, the activation of UCP1 acts as a small radiator in the brown and beige adipocytes [169,170]. Experiments performed with mitochondria from brown adipocyte have shown that free fatty acids can increase UCP1-mediated proton conductance. In these experiments it was observed that purine nucleotides can inhibit proton translocation by binding to the cytosolic face of UCP1. Under basal conditions, proton leakage is maintained by a predominant role of purine nucleotides on UCP1 [25]. Studies in rats observed that while the basal proton conductance represents 20 to 30% of the metabolic rate in hepatocytes, it increases to up to 50% in skeletal muscle. Taking into account the large proportion of skeletal muscle and the high metabolic activity of the liver, the metabolic rate in a mammal at rest is governed by proton conductance under thermoneutrality conditions and in the postabsorptive state [171]. However, experiments both in vivo and in cell cultures have shown that the thermogenic capacity of beige and brown adipocytes can be subject to external stimuli. In these experiments, it was observed that exposition of brown and beige adipocytes to cold induced a strong increase in mitochondrial respiration [8]. In humans, cold stimulation activates cold-sensitive thermoreceptors in the skin or viscera and transmits afferent signals to the hypothalamus and brain stem. Centrally, the release of noradrenaline from sympathetic nerves is stimulated, leading to the stimulation of postganglionic sympathetic nerves supplying brown adipocytes [105,172,173]. Noradrenaline acts on β-adrenergic receptors on the adipocyte surface, which generate a chain of stimulation that causes liberation of free fatty acids from stored triglycerides. Upon adrenergic stimulus that results in the activation of the brown (and white) adipocyte lipolytic cascade, respiration increases in a UCP1-dependent manner [174]. However, thermogenesis has been shown to be independent of lipolysis, and it has been demonstrated that stimulating lipolysis of cytosolic lipid droplets in brown adipocytes is not required for cold-induced non-shivering thermogenesis [175]. Recently, it has been observed in adipocytes that genetic or pharmacological elevation of levels in reactive oxygen species (ROS) is sufficient to drive thermogenesis [21]. Furthermore, studies in mice showed that application of heat stress (4 °C) or a β-adrenergic stimulus induces the activation of thermogenesis in BAT and results in an elevation of mitochondrial superoxide, mitochondrial hydrogen peroxide and lipid hydroperoxides. Oxidation of cysteine thiols by ROS can initiate the thermogenic activity of mitochondria in brown and beige adipocytes in a UCP1-dependent manner [176,177]. Similarly, it was identified that the accumulation of succinate, an intermediate metabolite of the tricarboxylic acid cycle in the mitochondria, rises independently of adrenergic stimulation in brown fat cells, and was sufficient to increase thermogenesis [178]. In this study, it was observed that the oxidation of succinate by the enzyme succinate dehydrogenase induces the production of ROS and manages thermogenic respiration [179,180]. The various pathways that thermogenesis may have in adipocytes were studied in UCP1 knockout mice (UCP1-KO). It was observed that UCP1-KO can be sensitive to cold, after being crossed with transgenic mice that express the PR domain containing 16 (PRDM16) that have the fatty acid binding protein 4 (Fabp4/aP2) promoter, which is expressed primarily in adipocytes [181]. Remarkably, UCP1-KO mice were resistant to diet-induced obesity at low temperatures, presumably by alternate activation pathways of energy loss, which are not yet well described. However, the possibility of thermogenesis being UCP1-independent remains controversial, and it is likely that thermogenesis in mice with UCP-1KO is induced by muscular activity that promotes shivering thermogenesis [182]. Creatine has been shown to be involved in metabolism and mitochondrial heat production, with recent observations suggesting the existence of a mitochondrial substrate cycle that is regulated by creatine to drive thermogenic respiration [176,183,184]. The thermogenic activity of creatine appears to only occur when ADP is limiting, which is expected during this physiological cellular state. However, the mechanism by which creatine influences the mitochondrial metabolism has yet to be established. Experiments using different animal models with genetic modifications have shown that reducing creatine may predispose animals to obesity. Interestingly, in a recent analysis of 18F-FDG PET/CT scans in human subjects, it was shown that renal creatinine clearance is a good predictor of activated human BAT [185]. Based on these observations and the fact that creatinine is a metabolite of the muscle energy store phosphocreatine, we can infer that creatine can be an activator of thermogenesis of BAT in humans and creatinine could be used as a biomarker of BAT activity [186].

## 6. Transcriptional Control of Browning

The phenotypic change in WAT into more energetically active cells indicates the involvement of gene expression regulation, in which internal or external regulators of WAT physiology must modify the chromatin structure or the DNA promoter methylation pattern of the target genes [74,187,188]. Recently, modifications of noncoding RNAs have been shown to act as an additional level of gene expression control [189,190]. Notably, a number of regulators function by modifying four transcriptional or coregulator factors: PPARγ, CCAAT enhancer binding protein beta (C/EBPβ), PPARγ co-activator-1α (PGC1α) and PRDM16. PPARγ and C/EBPβ act as transcription factors and directly bind DNA [191,192]. PRDM16 and PGC1α function as transcriptional coregulators, with PRDM16 forming a transcriptional complex with the canonical DNA binding transcription factors PPARγ and C/EBPβ through its zinc finger domains to activate the selective gene program for browning [193,194]. Similarly, it was observed in an analysis of chromatin immunoprecipitation sequencing (ChIP-seq) that PRDM16 co-localized with PPARγ and C/EBP in a large number of genes binding sites, further supporting their co-regulatory functions [195]. Although still not completely defined, beige adipocytes can be detected after the generation of a pre-adipocyte population of cells that are positive for platelet derived growth factor receptor-alfa (PDGFRα) and stem cell antigen 1 (SCA1) or precursors of MYH11. Its appearance occurs as a response to a variety of internal or external stimuli, including chronic exposure to cold, PPARγ agonists, cancer cachexia, exercise and various endocrine hormones [153,154]. Some factors that control the differentiation of BAT adipocytes also regulate the differentiation of beige adipocytes. Early beta-Cell transcription factor 2 (EBF2) is a key factor for the differentiation of BAT and has an important role in inducing the development of beige adipocytes [147,196]. EBF2 is highly expressed in PDGFRα positive cells, and the overexpression of EBF2 in primary white adipocytes or WAT induces the expression of thermogenic genes, increases oxygen consumption and suppresses high-fat diet-induced weight gain [197]. There are several proteins that can control beige differentiation through the functional control of PRDM16. The differentiation of beige adipocytes can be promoted by the activating or repressive activity of PRDM16. The formation of a repressor complex of PRDM16 with CtBP1 and CtBP2 reduces WAT adipogenesis [127]. On the other hand, the family of retinoblastoma proteins (pRb) antagonize the activity of PPARγ and PRDM16 [198]. pRb is a determinant of the choice of mesenchymal cells towards the osteoblastic lineage, thus in vivo experiments have shown that a pRb deficiency increases the development of mesenchymal precursor cells towards the brown adipocyte lineage. [199]. We previously observed that pRb inhibition by EP300 interacting inhibitor of differentiation-1 (EID-1) can induce the differentiation of beige adipocytes in humans [112]. Additionally, EID-1 can control adipogenesis through the transcriptional regulation of glycerol-3-phosphate dehydrogenases (GPDH), a key enzyme in the synthesis of triglycerides [200].

Several studies revealed peroxisome proliferator activated receptor γ coactivator 1 alpha (PGC1α) can regulate thermogenesis by directly inducing the expression of UCP1. PGC1α was first discovered as an interacting partner of PPARγ in brown adipocytes [201]. Pgc1α gene expression is greatly induced by cold exposure and is further activated following phosphorylation by the cAMP-PKA-p38/MAPK signaling pathway. PGC1α increases the transcription of specific genes through coactivation of by binding to transcription factors belonging to the nuclear receptor superfamily and recruitment of histone acetyltransferases such as CBP/p300 and GCN5 [202]. PGC1α binds to nuclear respiratory factors 1 and 2 (NRF-1 and NRF-2) to promote the activation of many mitochondrial genes. PGC1α mainly co-activates the nuclear hormone receptors, including PPARγ, PPARα, and estrogen related receptor (ERRα/β/γ), all of which participate in the transcription of brown fat genes [203]. PGC1α overexpression in adipocytes, myotubes, or cardiomyocytes promotes mitochondrial biogenesis and increases oxygen consumption [204,205]. Although PGC1α is a regulator of UCP1 expression, BAT Pgc1α-deficient mice display mildly increased lipid droplet accumulation but express normal levels of Ucp1 and other brown fat-selective genes [206]. Pgc1α-deficient BAT in culture fails to efficiently activate the thermogenic machinery in response to adrenergic stimulation [207]. These results demonstrate that PGC1α is required for the acute transcriptional activation of thermogenesis. Interestingly, the deletion of Pgc1α in adipocytes severely impairs the development of beige adipocytes in WAT [208].

## 7. Therapy with Inductors of Beige Cells

Every day the search for factors that can generate a metabolic change in the body is steadily expanding. Some of these experimental drugs have been used as possible obesity therapies by regulating BAT activity or adipogenesis of beige adipocytes [209,210]. Some regulatory peptides may have an effect on BAT and browning by stimulating hypothalamic nuclei. Dopamine agonists can induce BAT activity by stimulation of dopamine receptors 2 (D2R) in the VMH nuclei of the hypothalamus. It has been observed that patients treated with cabergoline, a D2R agonist, for 12 months showed a reduction in body mass index (BMI)and body fat together with an increase in resting energy expenditure (REE) and an improvement in glucose and lipid metabolism [211].

Polyphenols are substances characterized by the presence of several phenolic rings. They originate mainly in plants, which synthesize them in large quantities, as a product of their secondary metabolism. Some polyphenols are essential for plant physiological functions, others participate in defense functions in circumstances of stress. Among the polyphenols, flavonoids, catechins, capsaicin, and resveratrol stand out [212]. Capsaicin and capsinoids can trigger the activation of BAT and the browning of WAT. Studies with animal models of obesity have shown that capsinoids exert their mechanism of action by selectively activating the channels of the transient receptor potential vanilloid 1 (TRPV1) [213,214]. The effect of amplification of TRPV1 channels by capsinoid treatment ultimately results in activation of the vagal afferent nerves that project into the VMH hypothalamus [215]. Therefore, the mechanisms by which capsinoids induce BAT activity involve activation of β-adrenergic receptors. Human studies have shown that capsinoids can exponentially increase BAT activity after exposure to cold [216]. Resveratrol in elevated doses can reduce weight by increasing the activity of sirtuina 1 (SIRT1), a histone deacetylase dependent on NAD+. SIRT1 increases the function of AMPK-PGC1α and can trigger browning [217,218]. Recent studies have observed that the function of resveratrol can be mediated by the modification of the intestinal microbiome. The action of resveratrol on the gut microbiome is mediated by three potential ways: it can alter the composition of obesity-related gut microbiota, improve gut function and barrier integrity, or undergo gut microflora mediated-biotransformation to active metabolites in the intestinal tract [219,220]. Other polyphenols have been shown to regulate the activity of BAT; however, these observations are preliminary and have not been shown to have clinical relevance [212]. Gut microbiota can be considered as a contributing factor to the pathophysiology of obesity and may have potential therapeutic implications. Obesity is associated with elevated levels of Firmicutes such as Ruminococcaceae and depleted levels of Bacteroidetes such as Bacteroidaceae and Bacteroides. After exposure to cold in obese patients, an increase in the Firmicutes Ruminococcaceae family has been observed [221]. In fact, it was associated with high levels of acetate in plasma and was positively related to the expression of PRDM16 [222]. In support of these findings, several studies have shown that acetate increases the activity of brown fat and induces the formation of beige adipocytes [223,224]. Given the evidence from some studies, it seems that the use of prebiotics could improve adaptive thermogenic capacity. However, assumed that it is a new field, more studies are needed that can evidence preclinical observations. Furthermore, the underlying mechanisms that explain the relationship between prebiotic supplementation and the induction of thermogenesis and browning of white adipocytes are still indeterminate. An important point of connection between thermogenic capacity and prebiotics is the production of secondary metabolites by resident bacteria after the fermentation of prebiotics. Thus, the profile of short chain fatty acids (SCFA) and secondary bile acids could be important metabolites derived from the metabolism of bacteria capable of connecting prebiotic supplementation with the regulation of thermogenic capacity in BAT and WAT [225]. In recent years, the development of incretin effect mediations, especially glucagon-like peptide receptor agonists (GLP-1R), has had a beneficial effect, not only for the control of glucose levels in the blood but also for the reduction in body weight. The most prominent effects of GLP-1R are the anorectic effect and the modification of gastric emptying [226]. However, the GLP-1R effect on BAT energy expenditure, and possibly on WAT browning or hepatic lipid oxidation, will lead to a reduction in the weight and depletion in endogenous lipid stores [55]. The most likely effect of GLP-1R is mediated by modulation of AMPK activity located in the VMH nucleus of the hypothalamus. However, the possibility that extrahypothalamic areas are also involved in the effects of GLP-1R agonists on BAT thermogenesis and energy expenditure cannot be ruled out [227].

Considerable attention has been recognized to the secretory capacity of BAT and beige fat cells, and the substances secreted by these cells have been called “batokines”, which can have a paracrine or endocrine effect. These batokines play an important role in contributing to metabolic health by improving glucose and lipid homeostasis. The results of various BAT transplant studies have shown improvements in conditions associated with obesity, such as body weight and insulin sensitivity [228,229] (Figure 3).

The activation of β3-adrenergic receptors is perhaps the most widely used pharmacological method for the development of browning. Recently, a study of the β3-adrenergic receptor agonist mirabegron, an food and drug administration (FDA) drug approved for overactive bladder treatment, was used in patients with prediabetes. Treatment with mirabegron for 3 months decreased insulin resistance without causing a reduction in weight or any cardiovascular effects [231]. However, the development of adrenergic ligands for obesity and metabolic disease applications has the drawback of producing unwanted cardiovascular, autonomic, and bone effects over time [232,233,234]. Similarly, bone morphogenetic proteins 4, 7, and 8b (BMP4, BMP7, and BMP8-β), atrial and brain-type natriuretic peptides (ANP and BNP), FGF21, vascular endothelial growth factor-α (VEGF-α), and prostaglandins have all been shown to promote browning in vivo [89,235,236,237]. However, the use of these peptides as medications for obesity or metabolic diseases has induced undesirable effects that have restricted their clinical use. FGF21 is an activator of thermogenesis that has gained particular interest from various pharmaceutical companies. However, studies conducted in humans using FGF21 analogs have not shown a significant effect of these compounds on body weight or glycemic control [238,239], although a beneficial effect on hepatic steatosis has been observed [240]. Another experimental molecule as possible therapeutic target is SLIT2, a factor secreted by beige adipocytes. The expression of the SLIT2 gene is regulated by PRDM16. SLIT2 induces a PKA-dependent thermogenic pathway in adipocytes and improves general metabolic parameters in response to a high-fat diet. [128]. The c-terminal fibrinogen-like domain of angiopoietin-like 4 (FLD of Angptl4) induces cAMP-PKA-dependent lipolysis in white adipocytes and reduces diet-induced obesity [241]. Angptl4 increases the thermogenic program and promotes subsequent protection against weight gain and improved glucose tolerance in high-fat diet-fed mice [242]. Recently, kynurenic acid was observed to increase energy expenditure by activating G-protein-coupled receptor 35 (Gpr35), which in turn stimulates a thermogenic program in adipose tissue and increases regulator of G protein signaling 14 (Rgs14) levels in adipocytes, leading to enhanced β-adrenergic receptor signaling [243]. The results of many clinical studies in adult humans have suggested the beneficial effect of activating browning from the WAT. In an interesting study it was observed that a reduction in temperature (17 °C) for two hours a day for 6 weeks induced BAT activity and a progressive increase in energy expenditure [216]. Prolonged cold exposure for 5 to 8 h increased resting REE by 15%, increased glucose clearance in brown/ beige adipocytes, and significantly increased whole body glucose clearance [244]. In subjects with type 2 diabetes mellitus, 10 days of cold acclimation increased peripheral insulin sensitivity by 43% [245,246]. In another study, cold exposure produced a translocation in transporter type 4 (GLUT4) and increased glucose uptake in skeletal muscle [247]. Adipose-derived stem cells can be induced to differentiate to beige adipocytes in many ways, and the results of a large amount of research in this area suggest that a substantial number of potential drugs will become available in the next few years [248,249,250,251,252,253,254]. Although many effects remains to be clarified, it is evident that the increase in the activity of beige adipocytes is a consistent mechanism to increase glucose metabolism, supporting its role in treating obesity and related metabolic disorders in humans, especially for type 2 diabetes mellitus [255], which has two key pathophysiological components: peripheral resistance to the action of insulin especially in muscular cells and adipocytes and reduction in insulin secretion in β-pancreatic cells. A change in the sensitivity to insulin mediated by an increase in the number of beige adipocytes can be a highly valuable therapeutic strategy [256,257]. Additionally, a reduction in body weight may lead to a reduced load on pancreatic activity. Experiments carried out in humans have shown that a reduction in glucose levels combined with an increase in insulin sensitivity can be obtained by inducing beige/brown fat [244]. Despite the encouraging observations, from a therapeutic point of view some biochemical aspects should be evaluated. First, it should be determined whether the metabolic effects of beige adipocytes are subject to increased UCP1 levels or if there are alternative metabolic pathways that could improve the condition of adipocytes [183]. The effects of beige and brown adipocytes, and the UCP1-independent pathways, on the metabolic benefits in glucose and lipids, should be determined [89]. Another important aspect that should be taken into account is the detection of the volume of beige adipocytes in organisms. Although F-FDG-PET has considerably improved, it is desirable to develop new tools or instruments that can quantify the amount of beige adipocyte tissue in the body. Due to the current lack of understanding of beige adipocyte biology, it is necessary to gain a comprehension of the relevant physiological conditions, the number of days that the cells can survive, and the elements necessary to maintain the functionality of these cells.

Finally, it is important to recognize the specific factors that could induce plasticity in adipocytes, which may allow white adipocytes to be converted into beige adipocytes. The browning process from both adipocyte precursor cells and white adipocytes may be desirable for weight reduction and for the treatment of metabolic diseases.

## Figures and Tables

**Figure 1 metabolites-10-00471-f001:**
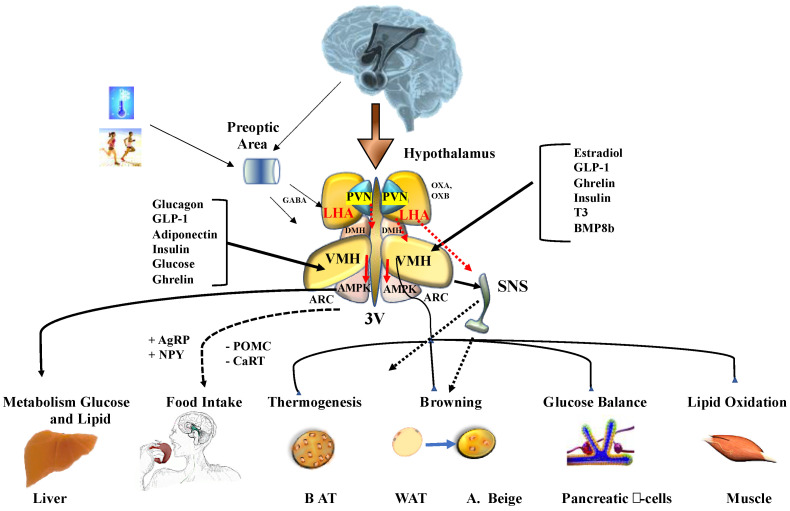
Energy homeostasis controlled by hypothalamus. Cold and physical peripheral signals reach the central nervous system where they interact with their specific receptors in the preoptic area. Adenine monophosphate activated protein kinase (AMPK) can regulate food intake, liver glucose production, lipid metabolism, brown adipose tissue thermogenesis (BAT), white adipose tissue (WAT) browning, and lipid and glycogen synthesis in skeletal muscle. AMPK activity in peripheral tissues is mediated by the activity of the sympathetic nervous system (SNS). Abbreviations: 3V, third ventricle; AgRP, agouti-related peptide; ARC, arcuate nucleus of the hypothalamus; BMP8b, bone morphogenetic protein 8B; CART, cocaine and amphetamine-regulated transcript; DMH, dorsomedial nucleus of the hypothalamus; GLP-1, glucagon-like peptide-1; LHA, lateral hypothalamic area; NPY, neuropeptide Y; POMC: pro-opiomelanocortin; PVN, paraventricular nucleus of the hypothalamus; T3, 3,3′,5-triiodothyronine; VMH, ventromedial nucleus of the hypothalamus. (+) increase intake; (-) reduce intake.

**Figure 2 metabolites-10-00471-f002:**
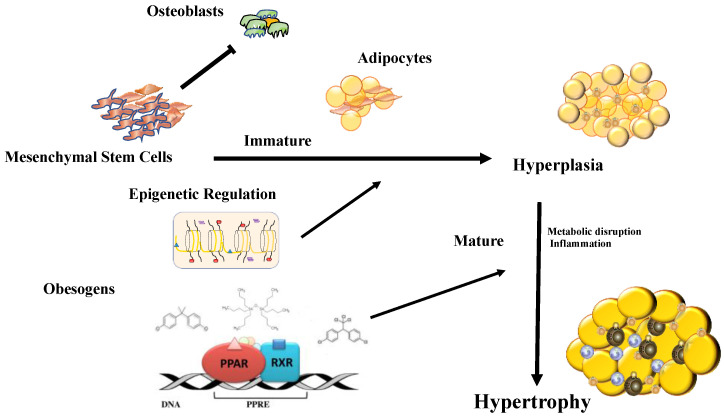
Chemical substances with the ability to modify thermogenesis primarily act through PPARγ:RXR receptors, the masters of adipocyte differentiation. Other changes that obesogenic substances can induce are epigenetic modifications (DNA methylation, covalent modifications of histones, and noncoding RNA). Obesogens can induce mesenchymal cells to direct differentiation towards the adipocyte line and reduce osteoblastic activity. Then, the obesogens induce hyperplasia of adipocytes and later produce hypertrophy with an accumulation of inflammatory cells that produce the secondary effects of obesity.

**Figure 3 metabolites-10-00471-f003:**
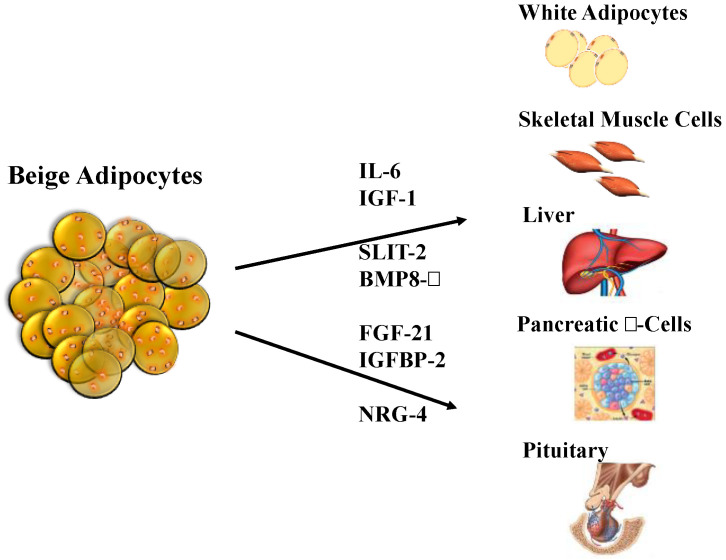
Adipokines secreted by beige adipocytes, which are also known as “batokines”. Beige adipocytes secrete molecules that have endocrine effects on some of the most important tissues involved in the regulation of body weight and lipid and carbohydrate metabolism. BMP8-β, bone morphogenetic protein 8b; IGF-1, insulin-like growth factor 1; IGFBP-2, insulin-like growth factor binding protein 2; IL-6, interleukin 6; NRG-4, neuregulin 4; SLIT-2, slit homolog protein 2. Adapted from Arroyave et al. [230].

**Table 1 metabolites-10-00471-t001:** Characteristics of different adipocyte tissues.

White Adipocytes (WAT)	Brown Adipocytes (BAT)	Beige Adipocytes * (Cold, TZD, FGF21, IL-4, IL-6)
Fatty Ac Oxidation (+)	Fatty Ac Oxidation (+++)	Fatty Ac Oxidation (+)	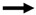	Fatty Ac Oxidation (+++)
Lipid Storage (+++)	Lipid Storage (+)	Lipid Storage (+++)	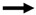	Lipid Storage (+)
Mitochondria (+)	Mitochondria (+++)	Mitochondria (+)	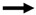	Mitochondria (+++)
PGC-1α (+)	PGC-1α (+++)	PGC-1α (+)	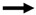	PGC-1α (+++)
Respiratory Chain (+)	Respiratory Chain (+++)	Respiratory Chain (+)	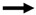	Respiratory Chain (+++)
Succinate (+)	Succinate (+++)	Succinate (+)	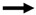	Succinate (+++)
UCP1 (-)	UCP1 (+++)	UCP1 (-)	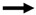	UCP1 (+++)
Unilocular	Multilocular	Unilocular	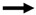	Multilocular
Markers: ASC-1, Resistin, Leptin	Markers: Eva1, Lhx8, Zic1	Markers: CD137, CIDEA, Cited, SLIT2, Tbx, Tmem26,

* Some conditions that can induce thermogenic activity of beige adipocytes. ASC-1, adipocyte-specific cell surface protein-1; CIDEA, cell death-inducing DFFA-like effector; CD 137, cluster of differentiation 137; Cited1, Cbp/P300-interacting transactivator 1; Eva1, epithelial V-like antigen 1; FABP4, fatty acid binding protein 4; FGF21, fibroblast growth factor 21; Lhx8, LIM/homeobox protein; PGC-1α, peroxisome proliferator-activated receptor-gamma coactivator alpha 1; Pdk4, pyruvate dehydrogenase kinase 4; SLIT2, slit guidance ligand 2; Tbx1,T-box transcription factor 1; Tmem2, transmembrane protein 26; TZD, thiazolidinedione; UCP1, uncoupling protein 1; Zic1, zinc finger protein 1. (+) increase; (+++) highest increase; (-) reduce; (
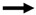
) Changes in Beige adipocytes after browning.

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
