# Peer review of "Control of Adipose Cell Browning and Its Therapeutic Potential"

_metabolites, 2020, doi:10.3390/metabo10110471_

Round 1

Reviewer 1 Report

This manuscript provides information related to energy homeostasis, different types of fat cells, adipocyte browning, and related potential therapeutics. The strength of the review is the vast amount of information related to the subject. The weakness is that not all contents seem necessary to draw out a concise conclusion.

1) Abstract 'During this process, changes in lipid inclusion occur, and the number of mitochondria~': it is not clear what it means by 'during this process'. The sentence just before includes temperature, nutrients, etc. but does not state which direction of each condition it means. Therefore, the two sentences does not seem to be connected well.

2) It would be either 'fat cell' or 'adipocytes'. 'Adipose cell' may infer other types of cells in adipose tissue. Is there any specific reason why the authors use this term?

3) Some sentences are difficult to understand exactly what it means and some grammatical errors and types are present. English editing by a professional service may be needed.

4) Symbols in figures are not correctly written.

5) The section on environment and obesity is interesting but it is not related to browning nor its therapeutic potentials, so it seems to diverge from the main focus of the review. I am not sure if this part is needed.

6) Among variations of fat cells, beige cells are most related to browning. However, the authors seem to have devoted the most effort on explaining white adipocytes (including full review on adipokines) and least for beige adipocytes. May be this is because of the following section on 5 and 6, in which case section 4 can be kept a little shorter.

7) Compared to the contents in section 1-6, the therapy section is rather weak. It would be great if the authors could classify the potential therapeutics in relation to what they have described in section 1-6 and expand more on the description for each candidate or include more candidates if possible.

Author Response

Reviewer # 1

This manuscript provides information related to energy homeostasis, different types of fat cells, adipocyte browning, and related potential therapeutics. The strength of the review is the vast amount of information related to the subject. The weakness is that not all contents seem necessary to draw out a concise conclusion.

  • Abstract 'During this process, changes in lipid inclusion occur, and the number of mitochondria~': it is not clear what it means by 'during this process'. The sentence just before includes temperature, nutrients, etc. but does not state which direction of each condition it means. Therefore, the two sentences do not seem to be connected well.

R/. Thank you for your comments, the manuscript was sent for grammatical corrections to MDPI.

  • It would be either 'fat cell' or 'adipocytes'. 'Adipose cell' may infer other types of cells in adipose tissue. Is there any specific reason why the authors use this term ?

R/. That is correct, we frequently use “adipocytes”. I will be check and perform the modifications.

  • Some sentences are difficult to understand exactly what it means, and some grammatical errors and types are present. English editing by a professional service may be needed.

R/. The manuscript was sent for a grammatical correction to MDPI

  • Symbols in figures are not correctly written.

R/. Thanks again for your observation. This was modified.

  • The section on environment and obesity is interesting but it is not related to browning nor its therapeutic potentials, so it seems to diverge from the main focus of the review. I am not sure if this part is needed.

R/. That is a point of discussion in the group if we must include this section in the manuscript. However, there are so many chemicals substances in the environment that may have an influence in the metabolism of WAT. In this regard, some polyphenols may have an important effect on BAT and Beige adipocytes. For this reason, it was included in the text of the manuscript.

  • Among variations of fat cells, beige cells are most related to browning. However, the authors seem to have devoted the most effort on explaining white adipocytes (including a full review on adipokines) and least to beige adipocytes. Maybe this is because of the following section on 5 and 6, in which case section 4 can be kept a little shorter.

R/. You’re right. However, information about white adipocytes is much higher than the Beige adipocytes. For this reason, one of the points that we want to point out is the plasticity of WAT. At some point, the browning of WAT must be the target of treatment in Obesity, because the volume of this tissue is larger.

  • Compared to the contents in section 1-6, the therapy section is rather weak. It would be great if the authors could classify the potential therapeutics in relation to what they have described in section 1-6 and expand more on the description for each candidate or include more candidates if possible.

R/. Thank you very much for your suggestion. We believe that the therapeutic section must be increased with the recent information and it was amplified in areas such as the effect of microbiota, glucagon-like peptides receptors agonists (GLP-1R), and some nutraceuticals.

Reviewer 2 Report

The manuscript is very hard to read, it needs a lot of work. Some sentences do not make any sense in english and, even though the authors cite and therefore seem to have read a good amount of the literature of they field, they make some really serious mistakes like stating

"..the brown adipose tissue (BAT) that has a high number of mitochondria and gives it the property of producing energy in the form of heat..."---> Not true, it's not about the number of mitochondria only but also the presence of UCP-1 (uncoupling)

"...BAT is only observed in the first few months of life" ---> Not true either, Ref.10 of the manuscript.

Due to those mistakes I would suggest to the authors to spend a lot more time reading the current literature and making the manuscript more reader-friednly (improve the english, make figures with bigger font and less blank space).

Author Response

Reviewer 2

The manuscript is very hard to read, it needs a lot of work. Some sentences do not make any sense in english and, even though the authors cite and therefore seem to have read a good amount of the literature of they field, they make some really serious mistakes like stating.

R/. The manuscript was sent for a grammatical correction to MDPI

"..the brown adipose tissue (BAT) that has a high number of mitochondria and gives it the property of producing energy in the form of heat..."---> Not true, it's not about the number of mitochondria only but also the presence of UCP-1 (uncoupling)

R./ You are right. UCP1 is the major effector that induce thermogenesis in the cells. However, recently a lot of work has shown UCP-1-independent thermogenesis in the mitochondria. It has been suggested that Ucp1−/− mice are protected from diet-induced obesity,  particularly at sub-thermoneutral temperatures, because they must induce alternative thermogenic pathways that are less efficient than the UCP1 pathway to sustain their body temperature. Consistent with this idea, Ucp1−/− mice can still activate ~50% of cold- mediated heat production compared with wild- type animals. However, just as  UCP1  is apparently equally thermogenic in beige adipose tissue and BAT52, there is no reason to assume that BAT cannot operate UCP1- independent thermogenic pathways, as has been proposed for beige fat.

- Kazak, L., Cohen, P. Creatine metabolism: energy homeostasis, immunity and cancer biology. Nat Rev Endocrinol 16, 421–436 (2020). https://doi.org/10.1038/s41574-020-0365-5.

- Antonacci, M. A. et al. Direct detection of brown adipose tissue thermogenesis in UCP1−/− mice by hyperpolarized (129)Xe MR thermometry. Sci. Rep.9, 14865 (2019).

- Meyer, C. W. et al. Adaptive thermogenesis and thermal conductance in wild- type and UCP1- KO mice. Am. J. Physiol. Regul. Integr. Comp. Physiol.299, R1396–R1406 (2010).

- Ukropec, J., Anunciado, R. P., Ravussin, Y., Hulver, M. W. & Kozak, L. P. UCP1-independent thermogenesis in white adipose tissue of cold- acclimated Ucp1−/− mice. J. Biol. Chem.281, 31894–31908 (2006).

- Granneman, J. G., Burnazi, M., Zhu, Z. & Schwamb, L. A. White adipose tissue contributes to UCP1-independent thermogenesis. Am. J. Physiol. Endocrinol. Metab.285, E1230–E1236 (2003).

- Ukropec, J., Anunciado, R. V., Ravussin, Y. & Kozak, L. P. Leptin is required for uncoupling protein-1-independent thermogenesis during cold stress. Endocrinology147, 2468–2480 (2006).

- Shabalina, I. G. et al. UCP1 in brite/beige adipose tissue mitochondria is functionally thermogenic. Cell Rep.5, 1196–1203 (2013).

- Kazak, L. et al. A creatine- driven substrate cycle enhances energy expenditure and thermogenesis in beige fat. Cell163, 643–655 (2015).

- Feldmann, H. M., Golozoubova, V., Cannon, B. & Nedergaard, J. UCP1 ablation induces obesity and abolishes diet-induced thermogenesis in mice exempt from thermal stress by living at thermoneutrality. Cell Metab.9, 203–209 (2009).

"...BAT is only observed in the first few months of life" ---> Not true either, Ref.10 of the manuscript.

R./ You are right. The BAT is observed in adults in a very small volume and the implications for physiologic are in study.

Due to those mistakes, I would suggest to the authors to spend a lot more time reading the current literature and making the manuscript more reader-friendly (improve the English, make figures with bigger font and less blank space).

R/. Thank you very much for your comment. As I wrote before the manuscript was sent for grammatical corrections. The figures were improved. The literature about BAT and Beige thermogenesis is so extensive that the purpose of the manuscript is not to be exhaustive nor is it intended to take up everything written about the field.

Reviewer 3 Report

This review addresses the mechanisms underlying adipocytes browning. This focus is of high interest in research as it may represent a tool for obesity treatment. Several studies and reviews have been provided on this topic, therefore the manuscript lacks novelty. However, the review provides a complete and exhaustive overview of the available literature and the conclusions are consistent with the topic of the arguments presented. The figures and references are adequate. However, English language needs extensive revision by a native speaker.

Author Response

Reviewer 3.

This review addresses the mechanisms underlying adipocytes browning. This focus is of high interest in research as it may represent a tool for obesity treatment. Several studies and reviews have been provided on this topic, therefore the manuscript lacks novelty. However, the review provides a complete and exhaustive overview of the available literature and the conclusions are consistent with the topic of the arguments presented. The figures and references are adequate. However, English language needs extensive revision by a native speaker.

R/. Thank you very much for your comment. The manuscript was sent for grammatical corrections. The figures were improved.  As you wrote the literature about BAT and Beige thermogenesis is so extensive that the purpose of the manuscript is not to be exhaustive nor is it intended to take up everything written about the field. However, we consider that in order to give a good message about the possibility of therapy for obesity by manipulating the thermogenesis of adipocytes, it is necessary to take into account several aspects of their physiology and molecular control.

Round 2

Reviewer 1 Report

The manuscript has been improved and is adequate in its present form.

Author Response

Thank you very much

Reviewer 2 Report

Even though the authors made some grammar corrections, the manuscript still needs extensive work, particularly in the introduction section. Some of the sentences state concepts that either are outdated or lack the proper reference.

“Furthermore, one of the difficulties in many countries is that obesity has not been declared a disease. For this reason, obesity is seen as a circumstance of the person and the risk of complications of obesity is not supported”.

https://www.ncbi.nlm.nih.gov/pmc/articles/PMC4988332/

Regarding my previous comments, they really haven’t addressed either of them.

1-"..the brown adipose tissue (BAT) that has a high number of mitochondria and gives it the property of producing energy in the form of heat..."---> Not true, it's not about the number of mitochondria only but also the presence of UCP-1 (uncoupling)

Authors response:

R./ You are right. UCP1 is the major effector that induce thermogenesis in the cells. However, recently a lot of work has shown UCP-1-independent thermogenesis in the mitochondria. It has been suggested that Ucp1−/− mice are protected from diet-induced obesity, particularly at sub-thermoneutral temperatures, because they must induce alternative thermogenic pathways that are less efficient than the UCP1 pathway to sustain their body temperature. Consistent with this idea, Ucp1−/− mice can still activate ~50% of cold- mediated heat production compared with wild- type animals. However, just as UCP1 is apparently equally thermogenic in beige adipose tissue and BAT52, there is no reason to assume that BAT cannot operate UCP1- independent thermogenic pathways, as has been proposed for beige fat.

The statement is about brown fat, not cells in general. Although others have shown UCP1 independent mechanisms like SERCA2b-mediated mechanism in beige fat (https://www.nature.com/articles/nm.4429.pdf?origin=ppub) or the creatine-mediated mechanism in an all fat- KO model like using the adiponectin-Cre driver (https://www.sciencedirect.com/science/article/pii/S1550413117304941?via%3Dihub), non of these mechanism have been validated in brown fat tissue, making their relevance in BAT yet unknown.

The current version it still says.“..the brown adipose tissue (BAT) that has a high number of mitochondria which provides it the property of producing energy in the form of heat…” .

This statement is at best inaccurate. If the amount of mitochondrial would be responsible for thermogenesis, then other tissues with a high amount of mitochondrial (e.g. the heart) would do thermogenesis as well.

2- "...BAT is only observed in the first few months of life" ---> Not true either, Ref.10 of the manuscript.

Authors response:

R./ You are right. The BAT is observed in adults in a very small volume and the implications for physiologic are in study.

Even though the authors acknowledge the mistake, not only they did not correct the statement in the manuscript, but also by neglecting to do so, it contradicts other sections of the manuscript.

However, BAT can be observed in adults in specific areas, such as the posterior neck and perirenal area [151-154]”.

Author Response

1- Thank you very much for your comments. The correction was made in the text of the manuscript.

2- Thanks again. We made the correction, pointing out the relevant fact of BAT in the production of heat in the first months of life in humans.

3- We try to improve the introduction of the manuscript

Reviewer 3 Report

After revision the text has significantly improved.

Author Response

Thank you very much

Round 3

Reviewer 2 Report

The authors addressed the suggested changes